# Preserved Sleep for the Same Level of Respiratory Disturbance in Children with Prader-Willi Syndrome

**DOI:** 10.3390/ijms231810580

**Published:** 2022-09-13

**Authors:** Qiming Tan, Xiao Tian (Tim) He, Sabrina Kang, Andrea M. Haqq, Joanna E. MacLean

**Affiliations:** 1Department of Pediatrics, Faculty of Medicine & Dentistry, University of Alberta, Edmonton, AB T6G 1C9, Canada; 2Women & Children’s Health Research Institute, Faculty of Medicine & Dentistry, University of Alberta, Edmonton, AB T6G 1C9, Canada; 3Stollery Children’s Hospital, Edmonton, AB T6G 2B7, Canada

**Keywords:** sleep-related breathing disorders, obstructive sleep apnea, polysomnography, growth hormone, before-after comparison

## Abstract

Debate remains as to how to balance the use of recombinant human growth hormone (rhGH) as an important treatment in Prader-Willi syndrome (PWS) with its potential role in obstructive sleep apnea. This single-center, retrospective study assessed differences in overnight polysomnography results between children with and without PWS and changes in respiratory parameters before and after the initiation of rhGH treatment in those with PWS. Compared with age-, sex-, and body-mass-index-matched controls (*n* = 87), children with PWS (*n* = 29) had longer total sleep time (434 ± 72 vs. 365 ± 116 min; *p* < 0.01), higher sleep efficiency (86 ± 7 vs. 78 ± 15%; *p* < 0.05), and lower arousal events (8.1 ± 4.5 vs. 13.0 ± 8.9 events/h; *p* < 0.05). Mean oxygen saturation was lower in PWS children (94.3 ± 6.0 vs. 96.0 ± 2.0%; *p* < 0.05), with no other differences in respiratory parameters between groups. Eleven children with PWS (38%) met the criteria for further analyses of the impact of rhGH; polysomnography parameters did not change with treatment. Compared with other children undergoing polysomnography, children with PWS had more favorable markers of sleep continuity and lower oxygen saturation for the same level of respiratory disturbance. rhGH administration was not associated with changes in respiratory parameters in PWS.

## 1. Introduction

Prader-Willi syndrome (PWS) is a multisystem neurodevelopmental disorder, with an incidence of 1/10,000 to 1/25,000 live births, caused by an absence of a functionally active paternal contribution in the chromosome 15q11.2-q13 region [1]. Clinical manifestations of PWS include infantile lethargy and hypotonia, contributing to poor feeding and failure to thrive, followed by excess weight gain and hyperphagia in early childhood, as well as hypogonadism, global developmental delay, intellectual impairment, and minor facial abnormalities. Management of PWS is multidisciplinary, with the goal of controlling weight, monitoring and treating comorbid conditions, and replacing hormone deficiencies, including recombinant human growth hormone (rhGH) [1]. Sleep disorders, including sleep-related breathing disorders, are common comorbid conditions in PWS [2].

Sleep-related breathing disorders in PWS present as early as infancy, with central sleep apnea (CSA) most commonly manifesting in infancy and obstructive sleep apnea (OSA) in childhood [3]. The prevalence of sleep-related breathing disorders in PWS is difficult to estimate as definitions of CSA and OSA vary across studies. While the prevalence of OSA in children with PWS is uncertain, it has been observed to be higher than in the general pediatric population, with OSA prevalence being as high as 80% in children with PWS of certain ages [4,5]. Sleep-related breathing disorders in PWS also show clinical characteristics that are different from those found in age- and body mass index (BMI) z-score-matched children without PWS [6]. Adenotonsillar hypertrophy, hypotonia, respiratory muscle weakness, pharyngeal narrowing, and hypothalamic dysfunction are important predisposing factors that may coexist in individuals with PWS [2]. Untreated OSA in PWS is associated with poorer neurocognitive outcomes and psychosocial deficits in children with PWS, who already face cognitive and behavioral challenges. Respiratory causes account for more than 50% of the deaths in children and adults with PWS, with some reported cases of sudden death occurring at night [7,8]. Therefore, early recognition and aggressive treatment of sleep-related breathing disorders are important to preserve cardiovascular health, improve daytime functioning, limit the risk of severe respiratory events during sleep, and improve quality of life in this population.

GH therapy is the only approved pharmacotherapy for PWS to date, with proven efficacy to normalize linear growth, significantly improve body composition and body mass index, and modify the natural history of PWS with a good safety profile [9]. Increasing evidence supports that rhGH therapy leads to better cognition and motor development in young infants and toddlers with PWS, while continued rhGH therapy into later childhood and adulthood may be beneficial for maintaining the improved body composition in adult patients [9,10]. Some early studies have also demonstrated favorable effects of rhGH therapy on sleep quality and respiratory function [11,12,13]. Unfortunately, several sudden death incidents during the initial phase of rhGH treatment, caused by suspected airway obstruction, were reported in the first few years after approval. rhGH therapy was thought to exacerbate underlying respiratory conditions in patients with PWS, which led to these incidents [14,15,16,17]. In a few cases, patients on long-term rhGH therapy had to discontinue the treatment due to the occurrence of severe OSA [18,19]. However, further studies revealed that PWS syndrome itself predisposes the individual to a risk of sudden death (with or without concurrent rhGH administration) [20,21]. This still raises a question about how to balance the use of rhGH as an important treatment in PWS with its potential role in OSA. The aims of this study were (i) to identify unique features in the clinical presentations of sleep-related breathing disorders and polysomnography (PSG) results of children with PWS and (ii) to determine what changes occur in these features after starting rhGH therapy.

## 2. Results

We identified 29 children with PWS who underwent PSG and matched them to 87 comparison children. Molecular genetics for the children with PWS showed deletion (*n* = 15, 52%), uniparental disomy (*n* = 7, 24%), imprinting defect (*n* = 3, 10%), and translocation (*n* = 1, 3%), with a specific defect not available for 3 children (10%). Children with PWS were matched with control children with respect to age, sex, and BMI z-score; children with PWS had lower height z-score (Table 1). Indications for PSG differed only related in relation to rhGH, with no difference in the frequency of other common indications. The only sleep-related symptoms that differed between children with PWS and matched controls were restless sleep and attention concerns, which were less common in children with PWS (Table 2). Within PWS, sleep symptoms did not differ by molecular genetics (data not shown).

Analysis of PSG variables from the initial study showed that children with PWS had longer total sleep time, higher sleep efficiency, and less arousals from sleep compared with the matched comparison group (Table 3). The proportion of sleep time spent in REM sleep did not differ by group. The only respiratory variable that differed between the groups was mean pulse oxygen saturation (SpO_2_), which was lower in PWS children. Hypoventilation, as measured by the percent time with end tidal carbon dioxide (ETCO_2_) > 50 mmgh (%TST with ETCO_2_ > 50 mmHg), did not differ between groups. There was no difference in the occurrence of OSA (PWS 59% vs. comparison 60%, OR 1.04, [95% CI: 0.55, 1.96]) or distribution of OSA severity between groups (PWS: no OSA 41.4%, mild OSA 31.0%, moderate OSA 17.2%, and severe OSA 10.3%; matched comparison group: no OSA 40.2%, mild OSA 33.3%, moderate OSA 10.3%, and severe OSA 16.1%; chi square 1.4, *p* = ns). On univariate analysis, the only demographic variable demonstrating a significant association with any OSA or moderate/severe OSA was boys having higher rates of any OSA (girls 48% vs. boys 73%, OR 1.56 [95% CI: 1.13, 2.15]) and moderate/severe OSA (girls 16% vs. boys 40%, OR 1.97 [95% CI: 1.15, 3.36]). Within the PWS group, molecular genetics were not associated with risk of any OSA (deletion: 59%, uniparental disomy 23%, translocation 0%, imprinting defect 0%, and not reported 18%; chi square 5.21, *p* = ns) or moderate/severe OSA (deletion: 62%, uniparental disomy 13%, translocation 0%, imprinting defect 0%, and not reported 25%; chi square 5.21, *p* = ns). Given the low number of children and uneven distribution of groups, this analysis could not rule out OSA risk difference by molecular genetic subtypes.

Of the 29 children with PWS, 24 (82%) had PSG prior to starting rhGH. Of these 24 children, 5 (21%) did not have follow-up studies, and 7 (29%) were started on noninvasive ventilation (i.e., continuous or bilevel positive airway pressure) before rhGH so subsequent studies were treatment studies. This left 12 children with PSGs before and after starting rhGH with 1 excluded as the follow-up study had a TST < 2 h, leaving 11 children with diagnostic PSG results before and after initiation of rhGH therapy. Overall, there was no change in any PSG parameters 6.8 (95% CI 2.0, 11.7) months after starting rhGH with a mean difference of 1.0 (95% CI 1.3, 0.80) years between PSG measurements (Table 4). Both increases and decreases in respiratory events were seen after the administration of rhGH (Figure 1). Of the six children that did not meet the criteria for OSA prior to initiation of rhGH, three met the criteria after starting treatment, with two having mild OSA (obstructive-mixed apnea-hypopnea index [OMAHI] 1.50 to 2.20 and 0.80 to 3.70 events/h) and one having moderate OSA (OMAHI 0 to 8.8 events/h), and three continued to not meet the OSA criteria with OMAHI < 1 event/h before and after treatment. Of the five children who met the criteria for OSA prior to initiation of rhGH, two did not meet the criteria after starting treatment (OMAHI 5.10 to 0.7 and 4.5 to 1.7 events/h) and three continued to meet the criteria for OSA (OMAHI 10.0 to 5.0, 3.1 to 8.8, and 4.40 to 10.7 events/h). None met the criteria for hypoventilation.

## 3. Discussion

The results from this retrospective review showed that children with PWS have more favorable sleep symptoms, with less restless sleep and attention concern reported by parents, and more favorable PSG sleep parameters, including longer total sleep time, higher sleep efficiency, and less arousals from sleep, compared with contemporary age-, sex-, and BMI-matched children who also underwent PSG. These differences are in the context of a similar index of respiratory events and carbon dioxide parameters with lower mean SpO_2_. PWS was not associated with an altered risk of OSA, suggesting that the risk of OSA and the severity of OSA are similar for PWS and other children undergoing PSG. Overall, being male sex was shown to be a risk factor for OSA with neither PWS nor BMI z-score altering this risk. Finally, PSG parameters overall did not differ before and after initiation of treatment with rhGH. Of note, just under one-third of children with PWS were started on noninvasive ventilation prior to starting rhGH, and one child developed severe OSA after starting rhGH.

Defining the risk of and risk factors for OSA in children with PWS is challenging because of differences in the definition of OSA, Refs. [22,23,24,25], with some studies reporting respiratory events from PSG without defining OSA, Refs. [26,27] as well differences in age and growth parameters. With no clear PSG standards for the identification of OSA in infants, OSA may be over-reported in infants if the pediatric criteria are applied to this younger age group [12,28,29]. Two studies reported rates of 27% and 44% based on obstructive AHI > 1.5 and >1 event/h, respectively, prior to rhGH therapy [24]. The findings of studies looking at risk factors for OSA in children with PWS are inconsistent, with some identifying association with older age, higher BMI, and adenotonsillar hypertrophy, while others failed to replicate these same findings [24,30]. In the present study, OSA incidence and severity did not differ from that of a matched comparison group. While PWS may confer a higher risk of OSA compared with otherwise healthy children, risk factors for OSA for PWS likely overlap with unselected children undergoing PSG. Further work is needed to understand OSA pathophysiology and its ensuing risk factors in children with PWS.

OSA is a multifactorial and heterogeneous disorder where airway compromise can be mitigated or worsened by an individual’s arousal response, upper airway response, and response to change in oxygen and carbon dioxide. Sleep in PWS has been characterized by a higher propensity for sleep, as measured by shorter sleep and REM latency, as well as longer sleep time on PSG, shorter mean sleep latency test scores, and excessive daytime sleepiness or hypersomnolence on validated questionnaires [31,32,33]. Studies measuring the arousal response have shown a higher arousal threshold to hypercapnia as well as abnormal arousal and cardiorespiratory response to hypoxia compared with controls, with a hypothalamic abnormality as a prime target to link hypersomnolence and abnormality in ventilatory control [34,35,36]. In the present study, we demonstrated that children with PWS had more favorable sleep symptoms and PSG sleep characteristics, including less restless sleep, longer TST, higher sleep efficiency, and lower arousal index compared with matched comparison children. This was despite similar respiratory event parameters and a lower mean SpO_2_. This suggests that children with PWS, for the same level of respiratory events, are less likely to experience arousal from sleep or may awaken from sleep later during respiratory events compared with other children with suspected OSA. Impairments in arousal or awakening responses, including responses to respiratory events or changes in oxygen and carbon dioxide, increase the risk of OSA even with a relatively uncompromised airway and may contribute to the risk of sudden death in the context of illness or other physiological disruption.

While there is controversy about a link between rhGH therapy and sudden death, there is widespread agreement on the benefits of starting rhGH treatment prior to obesity onset (typically by age 2 years), and emerging evidence supports very early initiation between 4 and 6 months of age in patients with PWS [37]. Reports of initial sudden death associated with rhGH therapy raised concerns over rhGH safety; postmortem findings suggested that the treatment might have led to OSA, respiratory infection, and sudden death in those patients [38]. In the largest study of adverse events in children with PWS receiving rhGH (N = 675), five children, aged 2.1 to 15.8 years, died, and two teens developed apnea [38]. In response, PSG has become a standard examination before initiating and during rhGH treatment. There are studies where symptoms of OSA developed or worsened during rhGH use in some children, though few ceased treatment or had dose reductions [24,39]. Several longitudinal studies, including the present one, demonstrated no association between rhGH treatment and increased risk of OSA in children with PWS [12,22,23,26]. Of note, there is no standard definition of what constitutes a clinically important change in OSA measurement; this may differ across the spectrum of mild-to-severe OSA as well as for different groups with OSA including PWS. Given that the risk of OSA during sleep in individual patients is multifactorial and changes in symptoms do not always reflect OSA, PSG before and after initiation of rhGH and if sleep symptoms worsen is prudent. For many years, as the only approved therapy for PWS, rhGH treatment has proven its long-term safety and efficacy in improving patients’ final height, body composition, and quality of life [37]. While OSA might be a risk factor for sudden death in children with PWS, it is not the only factor, as children without OSA experience sudden death as well [22,38,40]. Despite uncertainties, deciding against or delaying rhGH treatment because of the concern for rare serious adverse events must be balanced with the risk of developing complications in a child with PWS untreated with rhGH.

The unique cause of PWS may account for a component of the observed phenotypic variability in sleep problems in children with PWS. A large cohort study found that children with PWS had less parent-reported sleep problems, daytime sleepiness, and symptoms of sleep-disordered breathing than those with Rett and Angelman syndromes [41]. Similarly, in our study, children with PWS tended to have a less-worse sleep profile compared with their age-, sex-, and BMI-matched control subjects. Although we did not find any differences in sleep symptoms between molecular genetics, the potential direct and indirect contribution of PWS candidate genes to OSA and other sleep problems is worth further investigation. For instance, *MAGEL2*, an imprinted gene within the Prader-Willi critical region, has been shown to be important for the coordination of circadian rhythm in hypothalamic neurons [42,43,44], whereas circadian clock disruption might be a key contributing factor to OSA complications [45]. In addition, dysfunction of the autonomic nervous system is also present in patients with PWS [46]. There is evidence indicating a bidirectional relationship between sleep and autonomic activity [47]. Chronic sleep disruption impairs autonomic coordination, whereas the dysregulation of autonomic functions interferes with the initiation and maintenance of sleep. Future studies may consider exploring this bidirectionality in children with PWS, as it may play a crucial role in the pathophysiology of various sleep disorders, including OSA [47].

The limitations of the study must be acknowledged. As with any retrospective chart review, it is subject to selection bias and limited by the quality of the medical records. To maximize the numbers for a relatively rare condition, a broad time frame was used across which clinical practice, both with respect to PWS and OSA, had likely changed. To account for this in the comparison of PWS with a matched comparison group, children were matched for both demographic characteristics as well as the date of their PSG. In our study cohort, only one child with PWS and two children in the comparison group had a diagnosis of CSA, with no difference in the central index between the PWS and matched comparison group. With a relatively broad age range in our sample, further work is needed to understand the pathogenesis of previously described central sleep apnea associated with PWS. Additionally, the number of patients who had repeated PSG was small; thus, the statistical power to detect differences in PSG results before and after rhGH initiation was limited. A prospective trial comparing PSG results before and during rhGH treatment in a larger cohort of children with PWS across a broad age range is needed to confirm the results of the present study.

## 4. Materials and Methods

This study was a retrospective single-center chart review of children with PWS who underwent overnight PSG in a tertiary care pediatric sleep laboratory (Stollery Children’s Hospital, Edmonton, AB, Canada). Sleep laboratory records were screened to identify children who had undergone diagnostic PSG from January 2005 to September 2021; this time frame was chosen as PSG records were available from 2005. Health administrative data were also searched using International Classification of Disease codes for PWS, and this list was cross-referenced against sleep laboratory records. The final list was reviewed to confirm that all identified children had a genetically confirmed diagnosis of PWS. Each child with PWS was matched for age, sex, and BMI as well as date of diagnostic PSG, with three children undergoing PSG who did not have PWS. This matched comparison group did not exclude children with known risk factors for OSA or medical comorbidity. Data were extracted from sleep laboratory records and medical records. The study protocol and waiver of informed consent were approved by the University of Alberta human research ethics boards (Pro00064982).

Data collection included demographics, anthropometric measures, medical history, and PSG results. Height was measured using a stadiometer, and weight was measured on a digital scale. Height, weight, and BMI were converted to z-scores using available normative data [48]. Ethnicity was described by the parent/guardian. Presenting symptoms and medical history were extracted from medical charts and sleep laboratory questionnaires.

Diagnostic PSG was performed in accordance with clinical protocols at the time of the PSG, which included adherence to the standards of the American Academy of Sleep Medicine guidelines [49,50]. This included the determination of sleep state using electroencephalography, electrooculography, and submental electromyography. Channels to evaluate respiratory status included SpO_2_, nasal/oral air flow by thermistor, nasal pressure, and chest and abdominal wall movement using respiratory inductance plethysmography. CO_2_ was monitored using ETCO_2_ monitoring. Cardiac monitoring included electrocardiography.

Analysis of PSG data was completed by experienced scorers using the criteria of the American Academy of Sleep Medicine available at the time of the study [49,50]. Measures of sleep continuity included total sleep time (TST), sleep efficiency (percent time of time in bed spent in sleep), arousal index (number of arousals per hour of TST), and proportion of TST spent in stage 3 and rapid-eye-movement sleep (N3 and REM sleep, respectively). AHI was calculated based on the number of apneas and hypopneas during sleep divided by the total sleep time (TST). Oxygen desaturation index (ODI) was calculated based on the number of oxygen desaturation events ≥ 3% during sleep divided by the TST. An OMAHI index ≥ 1 events/h was used to define the presence of OSA [51]. The severity of OSA was further characterized as mild (OMAHI 1–4.9 events/h), moderate (OMAHI 5.0–9.9 events/h), or severe (OMAHI ≥ 10 events/h). As there are no clear parameters for the identification of OSA in infants, this pediatric OSA classification was used for infants as well to facilitate group comparisons, recognizing that this may over-represent OSA in children under 1 year of age [29]. Hypoventilation was identified by a CO_2_ > 50 mmHg for >25% of the TST. Studies were excluded from the analysis if the TST was <2 h.

The study data were managed using REDCap electronic data capture tools hosted by the Women & Children’s Health Research Institute, University of Alberta, Edmonton, AB, Canada. Statistical analysis was performed using the IBM© SPSS © Statistics (IBM Corp. Released 2019. IBM SPSS Statistics for Windows, Version 26.0. Armonk, NY, USA: IBM Corp). Student’s *t*-tests and Wilcoxon sign-ranked test were used to analyze paired and unpaired comparisons for parametric and nonparametric data, respectively. Categorical variables were analyzed using chi-square analysis, likelihood ratio (LR), or odds ratio (OR), as appropriate. A *p*-value < 0.05 indicated statistically significant effects. Binary regression analysis was used to assess potential demographic predictors of any and moderate/severe OSA, with covariates selection for multivariable analysis based on univariate analysis with *p* < 0.10.

## 5. Conclusions

This retrospective study found that, for the same level of respiratory events, sleep was preserved in children with PWS compared with other children, suggesting that arousal mechanisms may be altered in PWS and may contribute to the risk of OSA and sudden death. rhGH therapy was not associated with increased risk of OSA in children with PWS; this finding supports the recommendation that the treatment should be continued for as long as benefits outweigh risks. Future studies to better understand the pathophysiology of OSA and the consequences of OSA in PWS are needed. This includes establishing consistent definitions of OSA for children with PWS and evidence-based guidelines to support the management of OSA in this population.

## Figures and Tables

**Figure 1 ijms-23-10580-f001:**
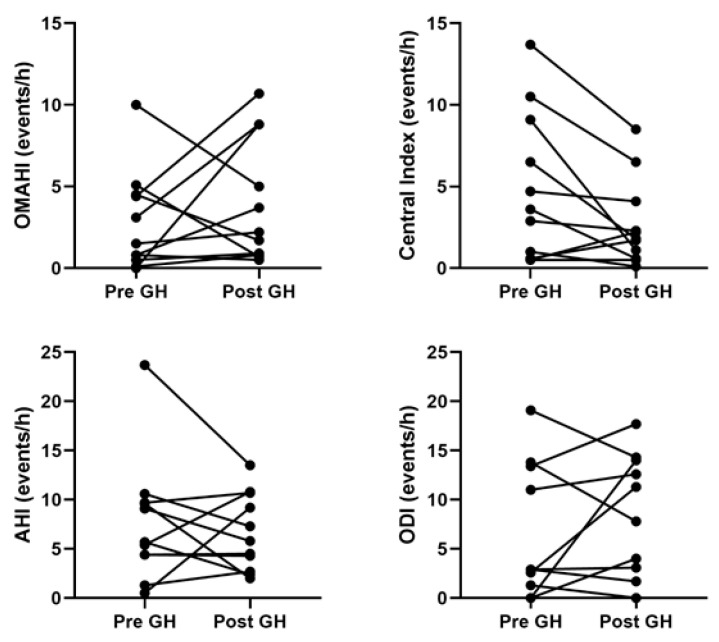
Analysis of summary polysomnography variables before and after starting recombinant human growth hormone showed no differences. AHI, apnea-hypopnea index; ODI, oxygen desaturation index; OMAHI, obstructive-mixed apnea hypopnea index.

**Table 1 ijms-23-10580-t001:** Demographic characteristics of children with PWS and the matched comparison group.

Parameter	Prader-Willi Syndrome (*n* = 29)	Matched Comparison Group (*n* = 87)
Age at PSG (years, mean ± SD)	4.4 ± 5.2	4.4 ± 5.1
Sex (F:M, % female)	17:12 (59%)	47:40 (54%)
Weight z-score ± SD	−0.99 ± 1.90	−0.12 ± 2.28
Height z-score ± SD **	−2.09 ± 1.99	−0.80 ± 2.08
BMI z-score	0.28 ± 2.28	0.38 ± 1.88
Indications for PSG (*n*, %) ^†^:		
Recombinant human growth hormone ^‡^	6 (21%)	0
Obstructive sleep apnea	22 (76%)	54 (65%)
Central sleep apnea	1 (3%)	2 (2%)
Excessive daytime sleepiness	1 (3%)	3 (3%)
Other	0	2 (2%)
Not reported	3 (10%)	5 (6%)

^†^ More than one indication for some children. ** *p* < 0.01, ^‡^
*p* < 0.0001. BMI, body mass index; PSG, polysomnography; recombinant human growth hormone, rhGH; F:M, female to male ratio.

**Table 2 ijms-23-10580-t002:** Sleep-related symptoms in children with Prader-Willi syndrome and matched comparison group.

Symptom	Prader-Willi Syndrome (*n* = 20)	Matched Comparison Group (*n* = 60)
Snoring	48%	47%
Witnessed apnea	14%	12%
Restless sleep **	14%	41%
Nighttime awakening	21%	28%
Morning headache	0%	11%
Mouth breathing	45%	52%
Daytime sleepiness	28%	27%
Poor school performance	21%	17%
Attention concerns *	0%	13%

* *p* < 0.05, ** *p* < 0.01.

**Table 3 ijms-23-10580-t003:** Polysomnography results for children with Prader-Willi syndrome and matched comparison group.

	Prader-Willi Syndrome	Matched Comparison Group
Total sleep time (min) **	434 ± 72	370 ± 118
Sleep efficiency (%) *	86 ± 7	78 ± 15
N3 sleep (%)	29 ± 15	31 ± 13
REM sleep (%)	27 ± 10	26 ± 13
Arousal index (events/h) *	8.1 ± 4.5	13.0 ± 8.9
AHI (median (IQR), events/h)	7.3 (11.8)	6.0 (13.8)
OMAHI (median (IQR), events/h)	3.2 (6.7)	2.5 (7.0)
Central index (median (IQR), events/h)	2.9 (10.1)	1.8 (5.4)
ODI (median (IQR), events/h)	3.6 (12.6)	4.7 (22.7)
Mean SpO_2_ (%) *	94.3 ± 6.0	96.0 ± 2.0
Minimum SpO_2_ (%)	82.9 ± 6.1	84.0 ± 11.1
%TST with SpO_2_ < 90% (%)	3.1 ± 7.3	4.2 ± 15.1
Mean ETCO_2_ (mmHg)	41.6 ± 58	41.5 ± 5.4
Maximum ETCO_2_ (mmHg)	51.9 ± 8.0	51.3 ± 8.8
%TST with ETCO_2_ > 50 mmHg (%)	3.1 ± 8.7	3.5 ± 14.5

* *p* < 0.05, ** *p* < 0.01 AHI, apnea-hypopnea index; CI, confidence interval; ETCO_2_, end tidal carbon dioxide; max, maximum; Max, maximum; Min, minimum; N3, slow wave sleep; ODI, oxygen desaturation index; OMAHI, obstructive-mixed apnea-hypopnea index; REM, rapid eye movement; SpO_2_, pulse oxygen saturation; TST, total sleep time. Data are presented as mean ± SD.

**Table 4 ijms-23-10580-t004:** Comparison of polysomnography results before and after initiation of recombinant human growth hormone.

Parameter	Before Starting rhGH	After Starting rhGH	Mean Difference (95% CI) or Wilcoxon Test Statistic
Age (years) ***	3.2 ± 3.3	4.3 ± 5.6	−1.0 (−1.3, −0.80)
Total sleep time (min)	467 ± 50	477 ± 38	−10.0 (−49.6, 29.3)
Sleep efficiency (%)	89 ± 5	88 ± 7	1.2 (−3.4, 5.8)
N3 sleep (%)	26 ± 15	29 ± 14	−2.9 (−17.3, 11.6)
REM sleep (%)	30 ± 13	22 ± 7	8.1 (−0.44, 16.2)
Arousal index (events/h)	11.4 ± 8.8	6.2 ± 2.5	5.2 (−1.2, 11.6)
AHI (median (IQR), events/h)	5.7 (5.3)	2.0 (0.0)	−0.58
OMAHI (median (IQR), events/h)	1.8 (3.5)	2.2 (8.0)	−1.07
Central index (median (IQR), events/h)	3.6 (8.5)	1.8 (3.5)	−1.89
ODI (median (IQR), events/h)	2.9 (13.4)	7.8 (12.3)	−1.58
Mean SpO_2_ (%)	93.5 ± 9.7	95.1 ± 1.9	−1.6 (−8.4, 5.1)
Min SpO_2_ (%)	85.0 ± 4.9	74.9 ± 23.1	10.1 (−4.6, 24.8)
%TST with SpO_2_ < 90% (%)	0.39 ± 0.35	3.3 ± 8.4	−2.9 (−8.5, 2.7)
Mean ETCO_2_ (mmHg)	39.7 ± 7.3	42.7 ± 2.7	−3.0 (−7.3, 1.4)
Max ETCO_2_ (mmHg)	48.8 ± 4.4	49.6 ± 7.1	−0.78 (−7.3, 5.7)
%TST with ETCO_2_ > 50 mmHg (%)	0.12 ± 0.18	2.0 ± 4.8	−1.8 (−5.1, 1.4)

*** *p* < 0.001 AHI, apnea-hypopnea index; CI, confidence interval; ETCO_2_, end tidal carbon dioxide; max, maximum; Max, maximum; Min, minimum; N3, slow wave sleep; ODI, oxygen desaturation index; OMAHI, obstructive-mixed apnea-hypopnea index; REM, rapid eye movement; SpO_2_, pulse oxygen saturation; TST, total sleep time. Data are presented as mean ± SD or as median (interquartile range).

## Data Availability

Data sharing is not applicable to this article.

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
