# Peer review of "Preserved Sleep for the Same Level of Respiratory Disturbance in Children with Prader-Willi Syndrome"

_ijms, 2022, doi:10.3390/ijms231810580_

Round 1
Reviewer 1 Report
I do not think that the article meets the aims and scope of the journal. This journal "provides an advanced forum for molecular studies in biology and chemistry, with a strong emphasis on molecular biology and molecular medicine."
My rejection of the manuscript is purely base on the discrepancy between the objectives of the articles and the journal's profile.
Reviewer 2 Report
I have read the article by Tan et al. with great interest. The authors analysed sleep physiology in patients with Prader-Willi syndrome and controls, in the former group also following growth hormone therapy.
Comments:
· Introduction. 2nd paragraph. Apart from CSA and OSA, I believe sleep-related hypoventilation syndrome is also prevalent in PWS (https://pubmed.ncbi.nlm.nih.gov/30007944/).
· Methods. I understand that you were limited in terms of the size of the PWS group. But how did you decide how many subjects to involve into the control group (it is 3-fold larger)?
· Results. Tables. Please, provide p values for comparisons.
· Results. GH is produced in N3 sleep. Did patients who require GH replacement have different sleep architecture?
Discussion. There were variable changes in respiratory parameters following GH therapy initiation. Sleep related respiratory events do show between-night variability anyhow. I wonder if you could discuss this, and what would you consider as a significant (beyond the expected variability) change in children.
Author Response
Responses to reviewers’ comments
The authors would like to thank the reviewers for their careful and comprehensive review of our manuscript titled “Preserved sleep for the same level of respiratory disturbance in children with Prader-Willi syndrome”. The manuscript has now been revised based on the reviewers’ comments. Please find below the changes and responses explaining point-by-point how each has been addressed.
Reviewer 1
I have read the article by Tan et al. with great interest. The authors analysed sleep physiology in patients with Prader-Willi syndrome and controls, in the former group also following growth hormone therapy.
Comments:
1. Introduction. 2nd paragraph. Apart from CSA and OSA, I believe sleep-related hypoventilation
syndrome is also prevalent in PWS (https://pubmed.ncbi.nlm.nih.gov/30007944/).
The review is correct that sleep related hypoventilation has been reported in children with PWS.
We measured this in our study as well (%TST with ETCO2> 50 mmHg) and found no difference between those with PWS and our matched control group (Table 3). We have added a sentence to highlight this (line 160).
2. Methods. I understand that you were limited in terms of the size of the PWS group. But how did you decide how many subjects to involve into the control group (it is 3-fold larger)?
While analyzing data from a rare population, the use of a lager control group is a common strategy to increase statistical precision. To adjust from possible confounders, we selected 3 matched comparison children for every child with PWS. This is included in our methods starting at line 87.
3. Results. Tables. Please, provide p values for comparisons.
P values that met our cut-off (p<0.05) are given below each table.
4. Results. GH is produced in N3 sleep. Did patients who require GH replacement have different sleep architecture?
This is an interesting question. This is outside the scope of this project so we cannot answer it with our data. We did report that N3 did not differ between the PWS and matched comparison group (included in Table 3) and that N3 also did not differ before and after starting rhGH (included in Table 4).
5. Discussion. There were variable changes in respiratory parameters following GH therapy initiation. Sleep related respiratory events do show between-night variability anyhow. I wonder if you could discuss this, and what would you consider as a significant (beyond the expected variability) change in children.
The issue of night to night variability relates to the challenge that there is no accepted change in OSA measurement that is consider clinically important. We have added a sentence to highlight this (line 271).

Reviewer 3 Report
Please see the attached review.

Round 2
Reviewer 1 Report
As previously mentioned, I do not think that the article meets the profile of the journal. Moreover, I do not understand the lack of need for an informed consent. Even with retrospective studies, authors still need to have an approval for patients enrolled in the study regarding the medical data usage.
Author Response
As previously mentioned, I do not think that the article meets the profile of the journal. Moreover, I do not understand the lack of need for an informed consent. Even with retrospective studies, authors still need to have an approval for patients enrolled in the study regarding the medical data usage.
- We requested a waiver of informed consent for this study as the study was retrospective with no contact with the participants. The request to waive the documentation of informed consent was reviewed and approved by the University of Alberta Research Ethics Boards. We have added the reason for the waiver to the manuscript.
Reviewer 3 Report
Please see the attached review.
